

# Non-linear genetic diversity and notable population differentiation caused by low gene flow of bermudagrass [*Cynodon dactylon* (L.) Pers.] along longitude gradients

Jing-Xue Zhang[1,2], Miaoli Wang[2], Jibiao Fan[1], Zhi-Peng Guo[2], Yongzhuo Guan[2], Gen Qu[2], Chuan-Jie Zhang[1], Yu-Xia Guo[2] and Xuebing Yan[1]

[1] College of Animal Science and Technology, Yangzhou University, Yangzhou, Jiangsu, China
[2] College of Animal and Veterinary Science, Henan Agricultural University, Zhengzhou, Henan, China

## ABSTRACT

**Background.** Environmental variation related to ecological habitat is the main driver of plant adaptive divergence. Longitude plays an important role in the formation of plant population structure, indicating that environmental differentiation can significantly shape population structure.

**Methods.** Genetic diversity and population genetic structure were estimated using 105 expressed sequence tag-derived simple sequence repeat (EST-SSR) loci. A total of 249 *C. dactylon* (L.) Pers. (common bermudagrass) individuals were sampled from 13 geographic sites along the longitude (105°57′34″–119°27′06″E).

**Results.** There was no obvious linear trend of intra-population genetic diversity along longitude and the intra-population genetic diversity was not related to climate in this study. Low gene flow ($Nm = 0.7701$) meant a rich genetic differentiation among populations of *C. dactylon* along longitude gradients. Significantly positive Mantel correlation ($r = 0.438$, $P = 0.001$) was found between genetic distance and geographical interval while no significant partial Mantel correlation after controlling the effect of mean annual precipitation, which indicated geographic distance correlated with mean annual precipitation affect genetic distance. The genetic diversity of *C. dactylon* with higher ploidy level was higher than that with lower ploidy level and groups of individuals with higher ploidy level were separated further away by genetic distance from the lower ploidy levels. Understanding the different genetic bases of local adaptation comparatively between latitude and longitude is one of the core findings in the adaptive evolution of plants.

# INTRODUCTION

Plants will undergo adaptive genomic and phenotypic changes in response to environmental heterogeneity on a spatial scale (*Yang et al., 2017*; *Rellstab et al., 2015*). Identification of adaptive genetic diversity and population structure of various populations of different

Corresponding authors
Yu-Xia Guo, guoyuxia@henau.edu.cn
Xuebing Yan, yxbbjzz@163.com

environments is of great significance for improving our understanding of the evolutionary process and exploring the adaptive potential of plants (*Li et al., 2017*). Landscape genetics is helpful for assessing the ability to produce and transmit adaptive genes (*Jordan et al., 2017*). Genetic differentiation and landscape genetic structure are the results of evolutionary processes including gene migration, random genetic drift and natural selection related to environmental variation due to latitude and longitude. However, gene flow among different populations can effectively prevent genetic difference caused by genetic drift (*Schoville et al., 2012*; *Rundle & Nosil, 2005*). Further study of the spatial patterns of genetic diversity is valuable in estimating evolutionary process and predicting the ability of populations to respond to climate change. Studying the genetic structure of more widely distributed plant populations would provide insights into genetic differentiation and evolution. Morphological plasticity and stress tolerances of utilized germplasm should match the environmental characteristics of a specific geographical region (*Casler et al., 2004*; *Boe & Casler, 2005*). Population distributions in a large scale could be shaped by regional climate and geographic factors such as latitude and longitude (*Peters et al., 2012*; *Sánchez-gonzález & López-mata, 2005*). Longitude is correlated with some environmental variables, and environmental factors (like temperature and precipitation) also play important roles in shaping these assemblages along the longitudes. Longitude was the driving factor behind the distribution of some study species, and the distribution of Carex physodes was controlled by longitude (*Zhang & Liu, 2016*). Decreasing MAT (mean annual temperature) and MAP (mean annual precipitation), along with increasing altitude, appear from the east to west along the longitude. Genetic isolation along longitude gradients caused by mountainous environments has been found in other widely distributed Europe-Asian plants (*Bartha et al., 2015*). Plants in eastern and western Greenland have been genetically different because the Greenlandic ice cap blocks gene flow (*Eidesen et al., 2013*; *Alsos et al., 2009*).

Bermudagrass (*Cynodon dactylon* (L.) Pers.) is a genus of the tribe Chlorideae (Poaceae), which inhabits most countries and islands, usually in between about 45°N and 45°S and penetrates to approximately latitude 53°N in Europe (*Burton, 1947*; *Casler & Duncan, 2003*). The warm season grass is widely used for turf and pasture because of its drought and heat tolerance and low maintenance requirements (*Harlan, 1970*; *Beard, 1972*). *C. dactylon* is an enormously variable and cosmopolitan warm-season turfgrass used for lawns, parks, and sports fields (*Shi, Ye & Chan, 2013*), which justifies new and innovative approaches toward evaluating and understanding the genetic relationships among different populations at different longitudes. Some studies have been used to analyze genetic diversity and population structure of bermudagrass because of its efficient dispersal and establishment abilities, rapid population growth, and phenotypic plasticity (*Linder et al., 2018*; *Zheng et al., 2017*; *Zhang et al., 2018*). Identifying the correlation between simple sequence repeat (SSR) markers and turf-quality-related traits among different bermudagrass populations can efficiently help to select high turf quality for bermudagrass. Different ploidy levels including diploid, triploid, tetraploid, pentaploid and hexaploid have been reported for *C. dactylon* (*Harlan et al., 1970*; *Wu & Taliaferro, 2009*). Polyploidy can improve competitive ability to adapt to new environments, which is vital for the worldwide distribution of grasses (*Te Beest et al., 2012*; *Linder & Barker, 2014*). There have

been multiple polyploidy events during plant evolution (*Wendel, 2015*), so evolutionary significance of polyploidy on wild plant populations is being studied. Polyploidy has been the subject of many studies for the genetic and genomic consequences of WGDs (whole-genome duplication) (*Soltis et al., 2016*).

The lack of genomic information makes genotyping difficult, but molecular markers enhance the study of plant genetic diversity (*Govindaraj, Vetriventhan & Srinivasan, 2015*). SSRs are characterized by high reproducibility, high polymorphism, co-dominant inheritance, and relative abundance across the entire genome (*Vasemägi, Nilsson & Primmer, 2005*), and are better selective markers for genetic population analysis of some orphan species (*Carneiro Vieira et al., 2016*; *Hodel et al., 2016*). EST-SSRs may reveal some different genetic patterns, because selective processes may affect their polymorphism (*Defaveri et al., 2013*). These molecular markers have great potential in genetic diversity and population genetic structure evaluation. Genetic diversity and population structure of 690 *C. dactylon* accessions have been analyzed using expressed sequence tag-simple sequence repeats (EST-SSRs), which is a crucial step of studying genetic evolution (*Vlachou et al., 1997*; *Jewell et al., 2010*). Although *C. dactylon* has important research value, the understanding of its population genetic structure and geographic variation is limited. Genetic diversity and genetic structure of *C. dactylon* along a latitudinal gradient across China and the relationship between genetic diversity and ploidy levels were also studied in our previous study (*Zhang et al., 2019*). Sampling includes maximum geographical variation from flooded east to arid west along the longitude to quantify the relative importance of longitude for adaptive genetic divergence of different bermudagrass populations. The genetic diversity of *C. dactylon* along longitude gradients may be correlated to complex geographic conditions and its biological characteristics including perennial growth, wind-pollination, polyploidy, sexual and asexual reproduction. Different environment factors along longitude gradients can also promote genetic variation and complex genetic structure of *C. dactylon*. Therefore, this study has four objectives (1) quantify population genetic diversity of 249 individuals from 13 populations ranging from different longitudes using EST-SSRs; (2) estimate the spatial population genetic structure and gene flow along longitude; (3) test the relationship between ploidy levels and genetic diversity; (4) compare genetic pattern between longitude and latitude. Studying genetic variation of the wild *C. dactylon* along longitude gradients can promote broadening the genetic resources, understanding the genetic process of adaptation to environmental variation along with longitude and providing some valuable information for the better utilization of wild germplasms of different ploidy levels in breeding.

## MATERIALS & METHODS

### Plant materials and DNA extraction

Plant materials consisting of 249 *C. dactylon* individuals were sampled at 13 different geographic origins locating 105°57′34″–119°27′06″E, China (Fig. 1). Mean annual precipitation and temperature at the collection sites were provided by the China Meteorological Administration (Table 1). Total genomic DNA was extracted using a

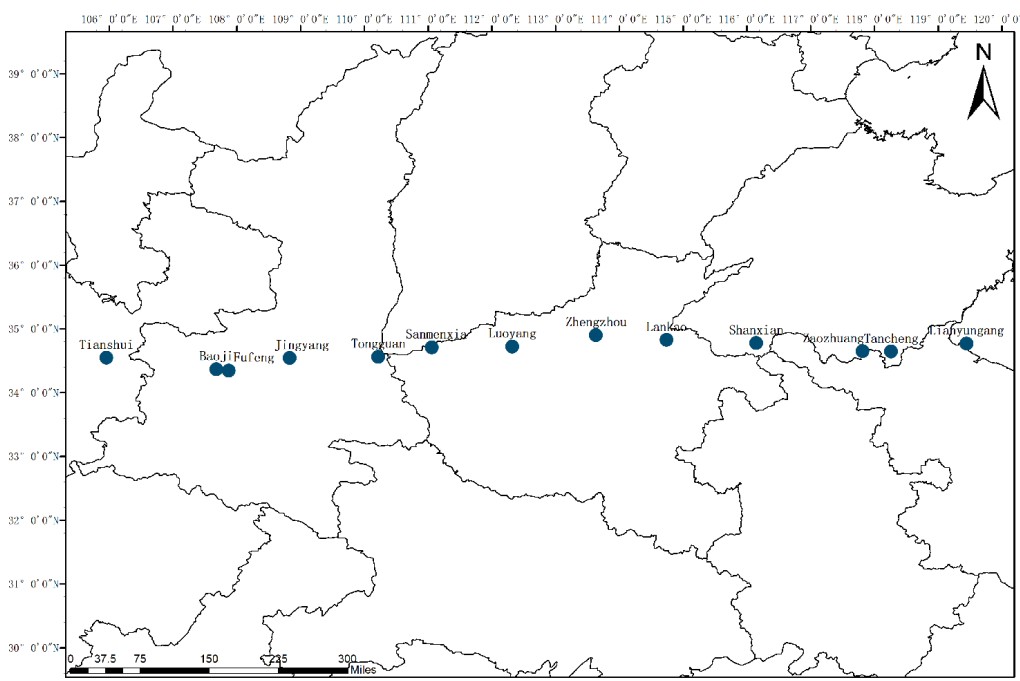

**Figure 1** **Locations of sampled populations of *C. dactylon*.** The blue dot represents the sampling sites along longitudinal gradient.

Zymo Research Plant DNA Extraction Kit (Xinmaijie, Zhengzhou, China) and the quality and concentration of DNA were checked by electrophoresis on 1% agarose gels.

## EST-SSR amplification

The 20 EST-SSR primer pairs from previous research (*Zhang et al., 2019*) produced clear and polymorphic bands and were subsequently used to evaluate genetic diversity for this study. 25 µL PCR amplifications reaction solution contains 1 µL of template DNA, 2.5 µL 10× PCR buffer, 2 µL of 25 mM $MgCl_2$, 0.5 µL of 10 mM deoxy nucleoside triphos phate (dNTP), 0.5 µL each of forward and reverse primers, 0.2 µL of Taq DNA polymerase, and 17.8 µL of nuclease-free water.

The amplifications were performed with the following PCR protocol used in the previous research (*Zhang et al., 2019*): first 1 cycle of 3 min at 95 °C, followed by 10 cycles at 95 °C, 60 °C and 72 °C respectively for 30 s, finally terminating with 1 cycle of 6 min at 72 °C. Next, the samples were subjected to 1 cycle at 95 °C for 3 min, then 20 cycles of 30 s at 95 °C, 55 °C, and 72 °C respectively, and finally 1 cycle at 72 °C for 6 min. PCR amplified fragments labelled with one fluorescent tag (6-FAM (6-carboxyfluorescein) (blue) and HEX (green)) were detected on a 3730XL Genetic Analyzer (ABI, USA) with the size standard GS500 LIZ. Capillary electrophoresis analysis was made by using GeneMapper Software Version 4.0 (Applied Biosystems, Foster City, CA, USA).

**Table 1** Information on geographical sites and climate of *C. dactylon* populations collected from different longitudes in China.

| Code | Region | Habitat | Longitude (E) | Latitude (N) | Altitude (m) | MAT (°) | MAP (mm) |
|------|--------|---------|---------------|--------------|--------------|---------|----------|
| 1 | Lianyungang | Island | 119°27′06″ | 34°46′09″ | 50 | 14.5 | 883.9 |
| 2 | Tancheng | Roadside | 118°16′08″ | 34°38′37″ | 30 | 13.8 | 832.9 |
| 3 | Zaozhuang | Roadside | 117°49′20″ | 34°38′48″ | 89 | 14.4 | 820.3 |
| 4 | Shanxian | Roadside | 116°09′11″ | 34°46′31″ | 30 | 14.2 | 621.4 |
| 5 | Lankao | Roadside | 114°44′55″ | 34°49′32″ | 60 | 14.3 | 631.1 |
| 6 | Zhengzhou | Swamp | 113°38′20″ | 34°54′04″ | 90 | 14.7 | 640.8 |
| 7 | Luoyang | Roadside | 112°19′30″ | 34°43′20″ | 210 | 14.4 | 637.2 |
| 8 | Sanmenxia | Roadside | 111°03′49″ | 34°42′29″ | 340 | 14 | 558.1 |
| 9 | Tongguan | Roadside | 110°13′18″ | 34°33′41″ | 540 | 13.1 | 602.9 |
| 10 | Jingyang | Roadside | 108°50′07″ | 34°32′32″ | 410 | 13.5 | 504.1 |
| 11 | Fufeng | Roadside | 107°52′41″ | 34°20′35″ | 570 | 12.8 | 569.9 |
| 12 | Baoji | Ditch side | 107°41′03″ | 34°21′54″ | 630 | 13.5 | 645.9 |
| 13 | Tianshui | Roadside | 105°57′34″ | 34°32′43″ | 1050 | 11.4 | 500.7 |

**Notes.**

MAT, mean annual temperature; MAR, mean annual precipitation.

## Analysis of genetic diversity

Genetic diversity for all populations and ploidy levels was calculated using POPGENE V. 1.32 Software (*Yeh et al., 1997*). These indicators included Nei's gene diversity index (He), Shannon diversity index (I) and polymorphic information content (PIC) (*Smith et al., 1997*; *Yeh, Young & Boyle, 1999*). He was generated according to genetic distance to analysis genetic diversity. I was computed based on the obtained allele frequencies, and PIC reflected the degree of variation of microsatellite DNA. Ploidy level of each sampled individual was determined by using a flow cytometer (Cube8, Partec, Germany) based on the method outlined by *Zhang et al. (2020)*. Linear association between ploidy levels and longitude was investigated under a regression analysis using the software package SPSS13.0 for Windows (SPSS Inc. Chicago, IL, USA) considering correlation coefficient values and significance of variables in the model. All individuals of the same ploidy level were treated as a population to compare the genetic diversity among different ploidy levels. Significant differences in genetic diversity parameters among different populations and ploidy levels were evaluated under a one-way analysis by Duncan's test using SAS 9.3 software (Institute Inc., Cary, NC, USA) (*Duncan, 1955*). We divided all the individuals into four groups by longitude from east to west to calculate the genetic diversity parameters for the four groups with four populations within one group as described above.

## Analysis of genetic relationship among all populations of *C. dactylon*

Distance matrices of intra-population genetic diversity indicators, environmental distance matrices, genetic distance matrices and geographical distance matrices based on the dataset for 13 populations were calculated using GenAlEx 6.1 (*Peakall & Smouse, 2006*). Mantel
correlation tests (*Mantel, 1967*) between intra-population genetic diversity indexes and environmental factors (temperature and precipitation) were conducted to examine the effects of environments on the intra-population genetic diversity. Mantel correlation coefficient between genetic distances and longitude distances for *C. dactylon* at different spatial scales was also investigated using NTSYS pc version 2.0 (*Khan et al., 2002*) and the probabilities for the significance were assessed using Bonferroni inequality adjustment. Partial Mantel test was conducted in R (ver. 4.0.5; *R Core Development Team, 2011*) to estimate correlation between genetic distances and geographic distance, while controlling for mean annual precipitation Analyses of molecular variance (AMOVA) and population differentiation index (FST) were estimated using ARLEQUIN v3.5 (*Excoffier & Lischer, 2010*). The significance of FST was determined by random resampling of the genotypic data through 1000 permutations. Principal coordinates (PCoA) analysis of genetic variations at different longitudes was performed using GenAlEx Version 6.1. With the help of clustering in NTSYSpc-v.2.1 software, UPGMA (unweighted pair-group method with arithmetic means) was used for cluster analysis to generate a dendrogram for genetic variation along longitude gradients (*Rohlf, 2000*).

## Population structure analysis

The population structure of the 249 individuals was constructed using a Bayesian clustering method implemented in STRUCTURE 2.3 (*Hubisz et al., 2009*) as previously described in *Zhang et al. (2019)*. The model-based approach implemented was used to subdivide the individuals into different subgroups. Specifically, twenty independent runs were performed for each simulated value of K ranging from 2 to 13. For each run, 10,000 iterations were set as the burn-in time following 100,000 iterations of Markov Chain Monte Carlo (MCMC). Outputs were processed with CLUMPP V1.1.2 (*Jakobsson & Rosenberg, 2007*) and STRUCTURE barplots were displayed using DISTRUCT 1.1 (*Rosenberg, 2004*). The best K value was determined from the modal value of the run with the delta K. Gene flow was calculated using POPGENE V. 1.32 Software (*Yeh et al., 1997*) based on the formula of Nm = 0.5(1- Gst)/Gst.

Mismatch distribution analysis (MDA; *Rogers & Harpending, 1992*) was used to investigate whether *C. dactylon* had undergone recent population expansion. Goodness-of-fit was tested with the sum of squared deviations (SSD) between observed and expected mismatch distributions (*Librado & Rozas, 2009*) and Harpending's raggedness index (*Harpending, 1994*) (HRag), using 1,000 parametric bootstrap replicates. The formula t $= \tau/2u$ was used to estimate the time since expansion began for the expanding population. The value u was calculated as u $= \mu kg$, where $\mu$ is the substitution rate ($6.5 \times 10^{-9}$ synonymous substitutions per synonymous site per year (*Gaut et al., 1996*)), $k$ is the average sequence length used for analysis (105 bp) and g is the generation time in years (1 year was used as an approximation for g according to age of first reproduction of *C. dactylon*). Fu's FS (*Fu, 1997*) and Tajima's D (*Tajima, 1989*) neutrality test were used to infer demographic history of each population. Statistical significance was estimated by performing 1,000 random permutations. Demographic analyses were performed in ARLEQUIN.

**Table 2  Within-population genetic diversity of 13 _C. dactylon_ populations at different longitudes.**

| Pop | N | He | I | PIC |
|-----|-----|--------|--------|------|
| 1 | 18 | 0.1939 | 0.2895 | 0.55 |
| 2 | 20 | 0.1367 | 0.2231 | 0.56 |
| 3 | 20 | 0.1157 | 0.1872 | 0.47 |
| 4 | 20 | 0.1837 | 0.2769 | 0.57 |
| 5 | 20 | 0.1843 | 0.2878 | 0.68 |
| 6 | 18 | 0.1859 | 0.279 | 0.53 |
| 7 | 19 | 0.0619 | 0.1131 | 0.40 |
| 8 | 16 | 0.1809 | 0.271 | 0.52 |
| 9 | 20 | 0.0984 | 0.1617 | 0.43 |
| 10 | 20 | 0.1338 | 0.2142 | 0.51 |
| 11 | 20 | 0.1603 | 0.2493 | 0.57 |
| 12 | 18 | 0.231 | 0.3408 | 0.62 |
| 13 | 20 | 0.1818 | 0.2779 | 0.57 |
| Total | 249 | 0.2587 | 0.4029 | 1.00 |

Notes.

He, Nei's gene diversity index; I, Shannon's diversity index; PIC, Polymorphic information content.

## RESULTS

### Genetic diversity and differentiation along longitude gradient

The genetic diversity parameters including the PIC value, He and I were calculated for EST-SSR markers in 13 populations (Table 2). The PIC values ranged from a high of 0.68 and a low of 0.40. Highest He value of 0.19 and lowest of 0.06 were calculated. The value of I ranged from 0.11 to 0.34 respectively for EST-SSR markers. Non-significant difference in genetic diversity parameters among groups (group 1 consisted of populations 1–3; group 2 consisted of populations 4-6; group 3 consisted of populations 7–9; group 4 consisted of populations 10–13) was observed along longitude gradients (He among populations: $F = 0.027$, $P = 0.974$). However, populations 4, 5, 6 and 8 at mid-longitude and populations 1, 12, 13 at lowest and highest longitude had higher genetic diversity than other populations according to I and He (Fig. 2). In general, there was also no significant correlation ($P > 0.05$) between intra-population genetic diversity and climate (mean annual temperature and precipitation), suggesting that temperature and precipitation showed little influence on intra-population genetic diversity along longitudes.

AMOVA results indicated that 38.23% of the genetic variation was attributed to difference among populations. About 61.77% of the genetic variation showed within population, which accounted for a greater part of the whole genetic variation (Table 3). Mantel's test revealed significantly positive correlation between genetic distance and longitudinal distance ($r = 0.276$, $P = 0.022$). Environmental measures along longitude gradients (geographical distance ($r = 0.438$, $P = 0.001$); mean annual precipitation ($r = 0.305$, $P = 0.033$)) were correlated with genetic distance, while mean annual temperature was not ($r = 0.006$, $P = 0.418$). However, a partial Mantel test accounting for mean annual precipitation was used to estimate that genetic distance was not independently correlated with geographic distance ($P = 0.146$). Gene flow (Nm) among these _C. dactylon_
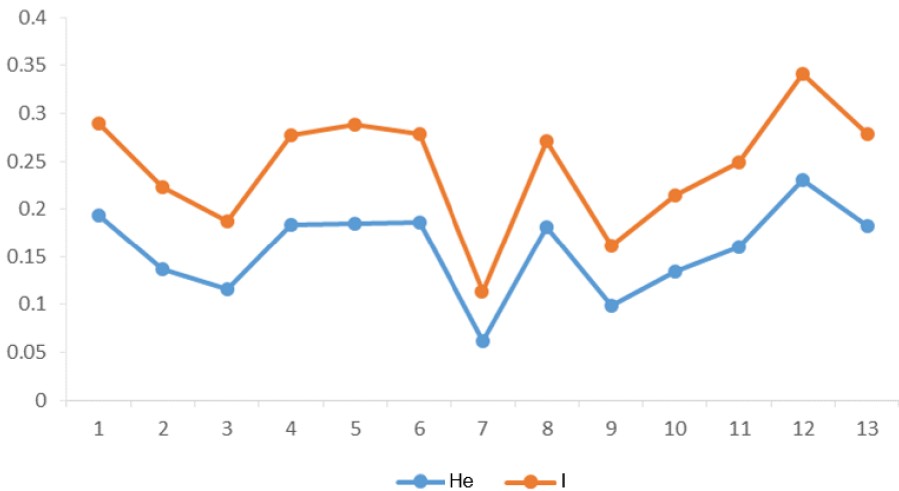

**Figure 2** **Summary statistics of within-population genetic variation of 13 *C. dactylon* populations from estern to western.** He, Nei's gene diversity index; I, Shannon diversity index.

**Table 3** **Analysis of molecular variance (AMOVA) of bermudagrass of different population.**

| Source of variation | df | Sum of squares | Variance components | Percentage of variation |
|---|---|---|---|---|
| Among populations | 12 | 1336.521 | 5.364 | 38.23% |
| Within populations | 236 | 2045.455 | 8.667 | 61.77% |
| Total | 248 | 3381.976 | 14.031 | |

**Notes.**
df, Degree of freedom

populations along longitude gradients was 0.7701 (<1), which mainly reflected low communication frequency of gene among populations.

## Genetic separation by UPGMA dendrogram and principal coordinate analysis

A UPGMA dendrogram dependent on Nei's genetic distance showed that 13 different populations were grouped into three distinct clusters (Fig. 3). The first cluster consisted of populations from Lianyungang and Tancheng that located at high-longitude in the east of China near the Yellow Ocean. Most populations along longitude gradient assigned to the second cluster, so a certain degree of kinship was identified among populations of *C. dactylon* located at different regions along longitude gradient. Populations 7, 9, 10 clustered together and population 3 was close to population 5. The third mixture cluster contained plants coming exclusively from interval locations along the longitude, such as Shanxian (4), Zhengzhou (6), Sanmenxia (8), and Baoji (12). In addition, each of four populations in the mixture cluster had higher intra-population genetic diversity. The Principal coordinate analysis (PCoA) (Fig. 4) revealed a similar pattern as inferred with UPGMA dendrogram. The first, second and third principal coordinates respectively explained 31.57%, 17.78% and 11.90%, altogether 61.25% of the variability. At axis one, 13 populations could be
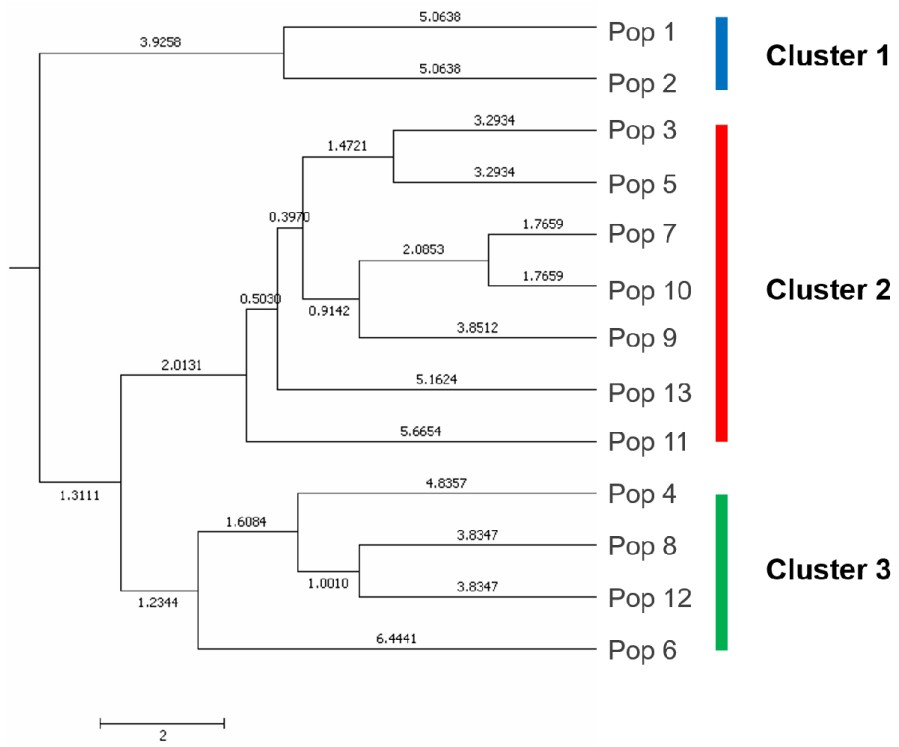

**Figure 3** **The UPGMA (unweight pair-group method with arithmetic means) dendrogram generated cluster analysis based on EST-SSR data of 13 _C. dactylon_ populations.** The _C. dactylon_ populations in this study were grouped into three clusters (blue, red, green).

divided into two parts with one of cluster 2 and others of clusters 1, 3. At axis two, cluster 1 consisting of populations 1 and 2 could be clearly separated from cluster 3. Additionally, populations of cluster 1 were clearly far from the other populations. Populations 3, 5, 7, 9 and 10 clustered together in the cluster 2. Populations 4, 6, 8 and 12 with relatively higher genetic diversity formed the cluster 3.

## Population genetic structure of _C. dactylon_

Bayesian analysis of the population structure of _C. dactylon_ was used to conduct separate calculation from $K = 2$ to $K = 20$. The results clearly demonstrated that the appropriate K values were obtained when the 13 populations were subdivided into 4 or 8 groups (Fig. S1). The results of 13 populations from K = 2–13 strongly indicated different clusters in populations of _C. dactylon_ (Fig. 5; Fig. S2). When $K = 4$, population 1 and 2 clustered together (blue) and the cluster with green consisted of populations 4, 6, 8, 12. Populations 5, 7, 10, 13 were differentiated from populations 3, 9, 11 compared to $K = 3$. And when $K = 8$, one group included populations 3, 5, 7, 9, 10, and another group consisted of the populations 8 and 12. Each of the other six groups consisted of only one population respectively, which indicated a rich genetic differentiation along longitude gradients. Admixture among populations of medium longitude (populations 5, 6, 8, 9) could be observed from the structure of _C. dactylon_ populations ($K = 2$ to 13), the clustering of

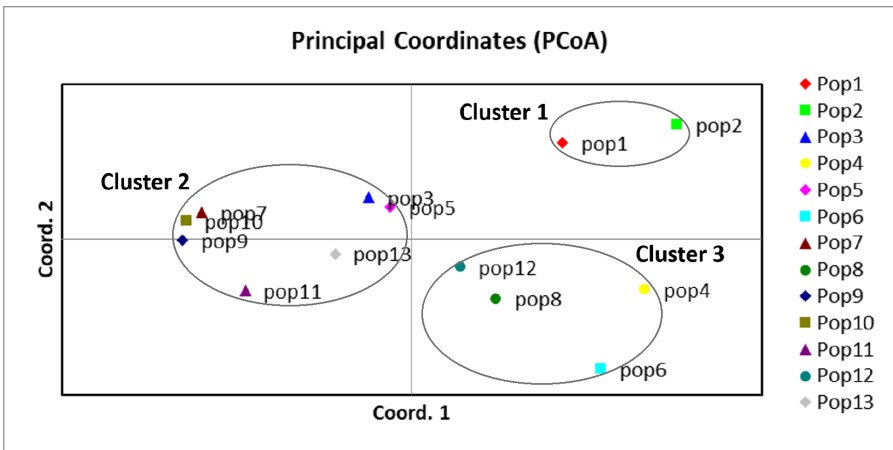

**Figure 4 Principal component analysis of the 13 populations based on the genetic distance derived from EST-SSR markers.** Populations in this study were grouped into three clusters (black circles).

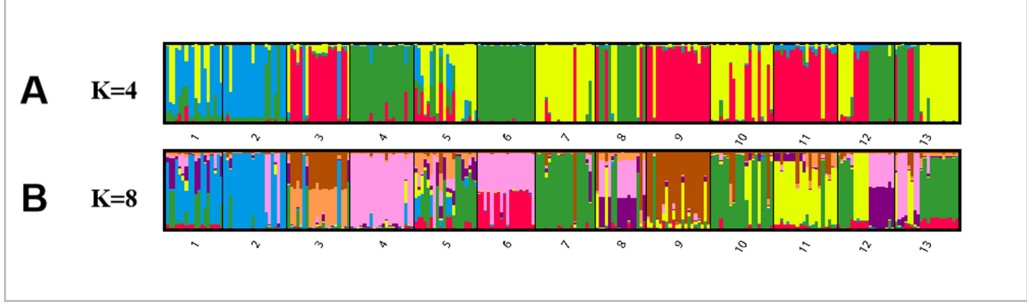

**Figure 5 Population structure analysis of 249 *C. dactylon* plants sampled from 13 populations using the model-based program STRUCTUREas follows: A-K = 4 and B-K = 8.** Each color shows one Bayesian ancestry group (K) and the length of the colored segment represents the estimated membership proportions in different Bayesian ancestry groups (K).

populations at low and high longitude were relatively stable except for the populations 1, 12, 13.

## Mismatch analysis and neutral test of *C. dactylon*

The mismatch distribution for each population of *C. dactylon* (Table 4) suggested populations 1–11, 13 possibly underwent recent population expansion. The result was also proved by significantly negative values of Fu's FS among the 13 populations of *C. dactylon*. Significant SSD values and non-significantly positive Tajima's D values of the population 12 might suggest relatively stable population size in Baoji. In addition, the demographic expansion time of each population was calculated based on the value of $\tau$.

## Genetic variation and differentiation among different ploidy levels

Genetic diversity (He and I) results of *C. dactylon* with five ploidy levels showed that triploid had significantly lower genetic diversity than pentaploid ($P = 0.019$ for He; $P = 0.008$ for

**Table 4 Results of the mismatch distribution analysis and neutrality tests of *C. dactylon* of each population along longitude gradients.**

| Populations | Tau ($\tau$) | Time since expansion began(t, Ma) | SSD | *P*-value | HRI | *P*-value | Fu's FS | *P*-value | Tajima's D | *P*-value |
|---|---|---|---|---|---|---|---|---|---|---|
| 1 | 29.500 | 21.612 | 0.006 | 0.960 | 0.013 | 0.820 | −5.704 | 0.017[*] | 1.152 | 0.916 |
| 2 | 0.000 | 0.000 | 0.013 | 1.000 | 0.017 | 0.940 | −9.177 | 0.000[**] | −0.370 | 0.377 |
| 3 | 2.000 | 1.465 | 0.008 | 0.870 | 0.011 | 0.810 | −10.399 | 0.001[**] | −0.299 | 0.435 |
| 4 | 16.000 | 11.722 | 0.005 | 0.390 | 0.009 | 0.610 | −7.288 | 0.002[**] | 0.813 | 0.838 |
| 5 | 24.500 | 17.949 | 0.010 | 0.650 | 0.020 | 0.280 | −7.270 | 0.007[**] | 0.072 | 0.559 |
| 6 | 21.200 | 15.531 | 0.008 | 0.540 | 0.011 | 0.880 | −5.904 | 0.013[*] | 1.116 | 0.888 |
| 7 | 0.400 | 0.293 | 0.012 | 0.700 | 0.019 | 0.800 | −14.636 | 0.000[**] | −1.731 | 0.033[*] |
| 8 | 30.400 | 22.271 | 0.023 | 0.340 | 0.020 | 0.670 | −4.805 | 0.017[*] | 0.944 | 0.881 |
| 9 | 2.600 | 1.905 | 0.011 | 0.390 | 0.009 | 0.830 | −11.688 | 0.000[**] | −0.570 | 0.301 |
| 10 | 8.700 | 6.374 | 0.008 | 0.540 | 0.011 | 0.680 | −9.330 | 0.002[**] | −0.116 | 0.495 |
| 11 | 21.500 | 15.751 | 0.004 | 0.900 | 0.007 | 0.960 | −8.116 | 0.004[**] | 0.194 | 0.619 |
| 12 | 35.300 | 25.861 | 0.021 | 0.000 | 0.020 | 0.500 | −4.927 | 0.020[*] | 1.493 | 0.940 |
| 13 | 28.900 | 21.172 | 0.009 | 0.780 | 0.016 | 0.560 | −7.349 | 0.002[**] | 0.762 | 0.840 |
| **Mean** | 17.003 | 12.457 | 0.011 | 0.620 | 0.014 | 0.718 | −8.199 | 0.007[**] | 0.266 | 0.625 |

Notes.
    SSD, sum of squared deviation under expansion model; HRI, raggedness.
    [*]Significant ($P \leq 0.05$).
    [**]Highly significant ($P \leq 0.01$).

I) and hexaploid ($P = 0.049$ for He; $P = 0.020$ for I) (Table S1). The genetic diversity of the *C. dactylon* individuals with higher ploidy level was significantly higher than that of individuals with lower ploidy level except for diploid. Significant genetic diversity exists among different ploidy level, but there is no significant difference in genetic diversity between pentaploid and hexaploid based on trend lines of PIC ($P > 0.05$). Diploid and triploid respectively had significantly lower genetic diversity than other polyploidy. All the individuals of *C. dactylon* at different ploidy levels fell into three major clusters based on UPGMA. The first and second clusters contained diploid and triploid individuals, respectively. Tetraploid, pentaploid, and hexaploid individuals formed the third cluster (Fig. 6). The genetic distance between two groups among different ploidy levels increased with the increase of ploidy level. In addition, there was no obvious growth trend of the distribution of ploidy levels, and regression analysis indicated that there was no significant linear association between ploidy level and longitude ($r = 0.071$; $P > 0.05$ –Fig. 7).

## DISCUSSION

### Genetic diversity and structure among populations along longitude gradients

Identifying relative roles of spatial and environmental factors in shaping patterns of population genetic differentiation is of interest for understanding landscape genetics. Each of 13 populations of *C. dactylon* along longitude gradients had a remarkable genetic variation within population based on the study. Genetic diversity can be improved by sexual propagation, and clonal propagation by rhizome and stolon can promote stable
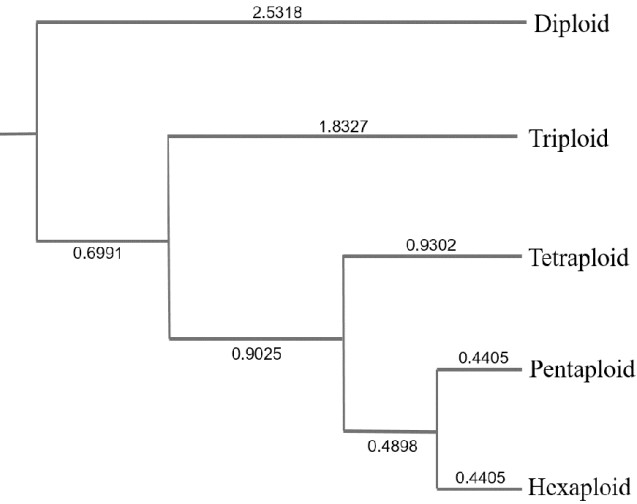

**Figure 6** The UPGMA (unweight pair-group method with arithmetic means) dendrogram of bermudagrass groups with different ploidy level.

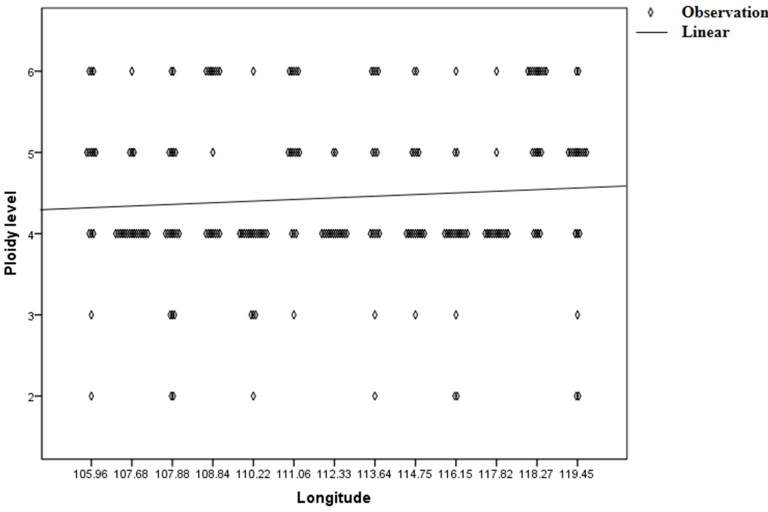

**Figure 7** Regression analysis between ploidy level of bermudagrass (*C. dactylon*) and longitude.

inheritance not being easily affected by genetic drift (*Loveless & Hamrick, 1984*). Intra-population genetic diversity had no obvious linear relationship with longitude in this study. In addition, mean annual temperature and precipitation at different longitudes had no significant influence on the intra-population genetic diversity. However, according to our previously published paper, higher within-population genetic diversity appeared at lower latitude with higher temperature (>25 °C), while relatively low temperature at
higher latitude might have little effect on the intra-population genetic diversity (*Zhang et al., 2019*). So higher temperature (>25 °C) may be associated with higher genetic variation of *C. dactylon*. From sampling regions, there was a little higher genetic diversity of populations at mid-longitudes (4, 5, 6, 8) than other populations at low- and high-longitudes, because outcrossing and self-incompatibility of *C. dactylon* could produce low communication among different populations of the east–west transitional locations. Genetic differences among distant geographical sites with unique alleles can increase genetic diversity (*Brochmann et al., 2003*; *Westergaard et al., 2011*; *Parducci et al., 2012*; *Eidesen et al., 2013*).

Landscape heterogeneity may lead to the formation of local adaptive populations, and different genotypes may be selected to adapt to different environments (*Felsenstein, 1976*; *Rieseberg & Willis, 2007*; *Anderson, Willis & Mitchell-Olds, 2011*). Environmental distance matrices were also calculated to estimate the influences of environmental distance matrices (IBE) on the genetic distance. No significant correlation was observed between genetic distance and geographic distance after controlling the effect of mean annual precipitation, which might be due to a correlation between geographic distance and mean annual precipitation. *P. arundinacea* (Phalaris) also exhibits a certain broad-scale population genetic structure in east-to-west gradients (*Perdereau et al., 2017*). The adaptive differentiation among populations is influenced by the natural selection of heterogeneous landscape and gene flow (*Dionne et al., 2008*; *Poelchau & Hamrick, 2012*). Low east-to-west inter-population gene flow (Nm = 0.7701) indicated that there was little adaptive selective pressure mediated evolutionary dynamics among different populations along longitude gradients (*Casler et al., 2007*). In addition, these factors to hinder further communication among populations include the transmission mechanism of pollen and seeds of species and the barrier caused by environmental conditions. In our study, restricted seed dispersal may exist among populations at different longitudes exhibiting significant IBD, and mountains in the Henan–Shanxi region may hinder further genetic communication among populations as natural barriers. Alpine plants from several mountain ranges in Europe usually have a strong patterns of longitudinal differentiation (*Ronikier, Cieślak & Korbecka, 2008*; *Ronikier, Schneeweiss & Schönswetter, 2012*). Geographical isolation along an ecological gradient or environmental isolation can lead to strong genetic differentiation for widespread plants (*Hoskin et al., 2005*).

The result of AMOVA indicated 61.77% of the genetic variation was attributed to differences within populations and FST was 0.3823. Higher FST along with increasing distance along longitude than latitude gradients may indicate local genetic differentiation among populations along longitude gradients (*Edillo et al., 2009*). Both spatial and environmental factors interweave in the formation of population differentiation pattern. In our case, low gene flow among neighboring populations may show the isolation-by-distance pattern among two neighboring populations. Several reproductive barriers in different environments within the same geographical area can block gene flow (*Lowry, 2012*; *Fishman et al., 2013*). Interestingly, some populations separated by 200 km in distance are close in genetic distance. A probable explanation is that human-mediated introduction allow contact among these populations. The mismatch distribution test and neutrality test
both indicated expansion within population along longitude gradients, with an estimated time around 12.5 Ma (under the demographic expansion model), which may be related to forests shrink and grasslands spread during Miocene time. The dramatic expansion of C4 grasslands in India is a part of increasing tropical C4 savannah vegetation worldwide in the late Miocene, which has been variously ascribed to uplift the Tibetan plateau (*Polissar et al., 2015*). In contrast, the population in Baoji with non-significant SSD and HRag values was interpreted as reflecting past demographic stability.

The low number of admixed genotypes at mid-longitude can be explained by the geographical distribution of the collection sites. Natural or human-mediated transmission may have increased genetic diversity and affected the genetic structure of several grass species (*Balfourier, Imbert & Charmet, 2000*; *McGrath, Hodkinson & Barth, 2007*; *Petit et al., 2003*; *Taberlet et al., 1998*). Hybridization at population boundaries and gene migration appear to promote homogenization. Populations at low and high-longitude may evolve much earlier than populations from sites of mid-longitude based on the result. Therefore, populations of *C. dactylon* at mid-longitude may evolve from complex introgressions between populations at low and high-longitude with close genetic affiliation. Historic changes in landscape configuration, spatial isolation and related gene flow changes may gradually lead to intraspecific genetic variation between eastern and western populations of *C. dactylon* in different geographical areas. The admixture among populations determines their potential to adapt to new environmental conditions (*Lee, 2002*).

## Relationship between ploidy levels and genetic diversity

The higher values of He, I and PIC were observed in populations with pentaploid and hexaploid than triploid. Polyploid retains most of the population's gene pool in the form of fixed heterozygosity (*Brochmann et al., 2004*). Hybridization of polyploid progeny from different parents of the same species can increase the genetic diversity. Multiple introductions from different origins and recombination between different genomes led to genetic mixture. Moreover, groups of individuals with the same ploidy level were separated by genetic distance along with the increasing ploidy levels. The higher genetic distance between two different ploidy levels along with the increasing ploidy levels was observed in the study, which indicated that each chromosome replication process might gradually increase the genetic dissimilarity. Diploid plants can participate in the formation of many polyploid (*Ramsey & Schemske, 2002*; *Otto, 2007*), and polyploid are often separated from their progenitors to produce new genotypes (*Čertner et al., 2017*; *Alsayied et al., 2015*; *Sweigart, Martin & Willis, 2008*). Different chromosome numbers could lead to reproductive segregation or sterility, which limited gene flow between hybrids and their parents (*Tate, Soltis & Soltis, 2005*). Polyploidy not only produce direct genetic redundancy, but also promote emergence of new gene functions and drive diversification and evolution (*Alix et al., 2017*). The hybrid genome composition of two related heterotetraploid species was analyzed using molecular cytogenetics to elucidate the role of heteropolyploid in the realization of *Hordeum* diversity (*Cuadrado, Bustos & Jouve, 2017*). From the facts above, polyploidization can lead to rich genetic variation and diversification for probable speciation of *C. dactylon*.

## Comparison of genetic pattern between longitude and latitude

No significant difference exists in genetic diversity parameters within-population along longitude gradients, while higher within-population genetic diversity appears at low latitude with higher temperature along latitudinal gradients (>25 °C) (*Zhang et al., 2019*). The result showed higher temperature might promote higher within-population genetic variation of *C. dactylon*. The barrier of hybridization of rice mainly focused on the environmental variation, just as temperature influenced cellularization of the endosperm to promote hybrid (*Folsom et al., 2014*). Higher temperatures may have affected the genetic similarity of *Culex pipiens* populations sampled early and later in the breeding season (*Edillo et al., 2009*).

A relatively low gene flow (Nm = 0.7701) means a rich genetic differentiation along longitude gradients (Nm < 1) (*Wright, 1965*). In our previous study on the genetic diversity and genetic structure of *C. dactylon* along a latitudinal gradient, low genetic differentiation and notable admixture structure existed among populations along latitudes (*Zhang et al., 2019*). Low genetic differentiation caused by different latitudes may be attributed to the wind-pollination outcrossing system and vegetative materials through human activities, which indicated that the latitude interval could not be an obstacle to genetic exchange. In addition, stronger admixture could be observed along with the latitude in another study (*Zhang et al., 2019*) than that along with longitude in this study, which indicated relatively lower gene flow among populations of *C. dactylon* occurred along the longitude gradients. North-south direction gene flow has existed between the eastern *Lemus chinense* populations and the western ones (*Long et al., 2019*). The eastern Eurasian populations of some species with global distribution pattern are a little different from-the western Eurasian populations, which occurs between late Miocene and Pleistocene (*Tabata et al., 2018*; *Hardouin et al., 2015*; *Rodrigues et al., 2014*). East Asia was less extensively glaciated and had a relatively mild glacial climate. The Qinghai-Tibetan Plateau uplifted rapidly in Pleistocene and the Asian summer monsoon intensified, so the formation of the inland arid belt of Eurasia resulted in long-term ecological isolation (*Clark & Mix, 2002*; *López-Pujol et al., 2011*). A probable explanation is that the genetic differentiation along longitude gradients was little and the gene flow was frequent before the uplift of Tibet Plateau, but the situation reversed after that because populations were ecologically isolated. Therefore making a comparison between latitude and longitude, a stronger gene flow from the north-south than east–west direction is characteristic of genetic structure of *C. dactylon* in China. Distinct genetic divergence of *C. dactylon* presented between the lower and the higher ploidy level along both longitude and latitude gradients. In addition, widespread polyploidization may cause significantly increasing genetic diversity with the ploidy level.

## CONCLUSIONS

A large-scale population genetic survey of genetic diversity of *C. dactylon* has been conducted to evaluate genetic variation and population structure along a longitude gradient. Landscape heterogeneity and geographical isolation along several mountain ranges can lead to formation of locally adapted populations with reduced gene flow between habitats.

Geographic distance correlated with precipitation may influenced genetic diversity among populations along longitude gradients. A higher level of genetic differentiation among populations of *C. dactylon* along longitude gradients than those along latitudinal gradients indicated a stronger gene flow from the north-south than east–west direction. Decreased genetic diversity is estimated for individuals with lower ploidy level compared to higher ploidy level, and relatively rich genetic differentiation exists among different ploidy levels. Identifying genetic diversity along longitude gradients in contrast with latitudinal gradients provides important genetic information and improves the ability to protect biodiversity.

### Funding

This work was supported by National Natural Science Foundation of China (Evolution of Cynodon dactylon along latitude and longitude indicates the genetic diversity, phylogeny and phylogeography). The funders had no role in study design, data collection and analysis, decision to publish, or preparation of the manuscript.

### Grant Disclosures

The following grant information was disclosed by the authors:
National Natural Science Foundation of China: Evolution of Cynodon dactylon along latitude and longitude indicates the genetic diversity, phylogeny and phylogeography.

### Competing Interests

The authors declare there are no competing interests.

### Author Contributions

- Jing-Xue Zhang conceived and designed the experiments, performed the experiments, analyzed the data, prepared figures and/or tables, authored or reviewed drafts of the paper, and approved the final draft.
- Miaoli Wang, Jibiao Fan, Zhi-Peng Guo, Yongzhuo Guan and Gen Qu performed the experiments, prepared figures and/or tables, and approved the final draft.
- Chuan-Jie Zhang performed the experiments, authored or reviewed drafts of the paper, checked linguistic accuracy, and approved the final draft.
- Yu-Xia Guo and Xuebing Yan conceived and designed the experiments, authored or reviewed drafts of the paper, and approved the final draft.

### Data Availability

The raw data is available in the Supplemental Files.

### Supplemental Information

Supplemental information for this article can be found online at http://dx.doi.org/10.7717/peerj.11953#supplemental-information.

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
