# Peer review of "Non-linear genetic diversity and notable population differentiation caused by low gene flow of bermudagrass [Cynodon dactylon (L.) Pers.] along longitude gradients"

_PeerJ, doi:10.7717/peerj.11953_

## Round 0.1 · original submission · Major Revisions

Both reviewers found this work to be of interest, but both highlighted real problems with the manuscript as presented. This will require substantial revision and possibly be sent out to review again. Please address the reviewers' concerns, with special focus on:

Clarity of writing and methods
Presentation of appropriate data (e.g. flow cytometry) and methods (scripts on GitHub or Figshare)
Reworking the intro/discussion to better frame the paper
Addressing Rev. 1 concerns about demography

Reviewer 1 ·

Basic reporting

1. This paper is difficult to read due to its many errors (grammar, verb tense, word choice, and incomplete/run-on sentences). I would classify the extent of these errors as severe, but with sufficient proof-reading and revision, they could be fixed. I mainly regret that I was unable to understand some key sentences, as their meaning was unclear. In some cases, I was unsure whether I was having difficulty with a sentence due to an error or because the authors were making a conceptually challenging point. There are too many mistakes to note, but I will draw attention to a small selection:
88: This sentence is incomprehensible.
234: I don't know if they mean "low temperature" or "low temperature variability."
243: The "because" doesn't seem to follow logically.
260: The phrase "no need" seems out of place here.

2. Several important concepts were not adequately discussed in the Introduction and Materials & Methods.
Crucially, the justification for their interest in longitude as a meaningful environmental gradient is not addressed in the introduction. Though many environmental variables predictably co-vary with latitude, covariance with longitude is case-specific, not general. Further, they do not justify their choice of environmental variables, though mean annual temperature and mean annual precipitation are frequently default selections. On line 226 in the Discussion, they begin to talk about the other environmental complexities that may account for the non-linear patterns of genetic diversity. In the next paragraph, starting at 244, the authors begin to discuss aridity, flooding, mountain ranges that hinder gene flow, and much later, they mention glaciation history. All of this information is greatly beneficial, and should be moved to the introduction (and expanded) as justification for this question. Even after this revision, the question would remain: Why structure your research question around longitude? The plant experiences myriad environmental variables, but longitude is a human construct of no direct biological importance (as far as I know, though if I am wrong, the authors should explain why). It seems that these same analyses could be re-contextualized in reference to temperature, precipitation, elevation, soil type, habitat fragmentation, or any combination of environmental variables, for a more intuitive and direct discussion of the association of genetic diversity, population structure, and environment in these sampled populations.
The role of demography in shaping geographic patterns of genetic diversity is neglected. Throughout the manuscript, I found myself wondering about the demographic history of these populations, their population sizes, and the direction and timing of dispersal. The paragraph beginning at 280 comes close to discussing these ideas, and with some revision, may provide some valuable insight for the paper.
They do not explain why their model system is suitable for their study questions. In particular, I wonder why you would choose a weedy, invasive species for investigations of local adaptation, a process which involves multiple generations of directional selection within an environment and which produces organisms particularly suited for their home environment. They could also be more clear on their criteria for population selection and specimen harvesting.

3. In general, sufficient literature seems to be referenced when discussing ideas. See below for some exceptions:
125: The three statistics for genetic diversity should be referenced with their corresponding citations.
268: This sentence is written as if the numbers are applicable across many systems, but in fact these numbers were from an experiment on mosquitoes and should not be generalized. As written, this sentence is highly misleading.
290: Generally accepted according to who, or in what circles?
291: It has been hypothesized by whom?

Experimental design

1. This manuscript meets the criteria for inclusion in this journal (original primary research within the scope of the journal). The research questions are defined, relevant, and meaningful, though they would be more impactful if the Introduction better explained their relevance, and they fill an identified knowledge gap.

2. However, this experiment has considerable statistical and conceptual flaws:
They utilized a simple Pearson correlation to establish a causal relationship between environmental conditions and genetic diversity. This approach disregards neutral effects due to demography (shared ancestry, gene flow). Numerous methods (BayEnv, Sambada, LFMM) have arisen to account for neutral effects when investigating gene/environment associations, and this principle should be applied here. A Pearson correlation is too weak a tool for addressing this research question.
They provide no way for the reader to know which populations are which ploidy level. I suspect that they considered and investigated this angle but failed to report it. This is an unfortunate oversight, as patternation of ploidy level across the longitudinal transect or composition of different ploidy levels within a population could explain (at least in part) the nonlinear patterns of population structure and/or genetic diversity. Not accounting for ploidy level when running a Pearson correlation between genetic diversity and environment would almost certainly confound results. Perhaps the effect of environment on levels of genetic diversity is present and significant, but is masked by the greater effect of ploidy differences.

3. At times, the authors apply analytical tools without justification.
They do not distinguish their three statistics of genetic diversity, nor do they explain why they used all three.
They cluster populations using both UPGMA and PCoA (Figures 3 and 4). Both provide the same clustering results. Why use both?

4. The authors provide a fair amount of detail in methodology, except for the few cases I have pointed out previously. I would have preferred to be able to see the scripts that were used in data analysis, but these were not provided.

Validity of the findings

1. I am suspicious of some of the conclusions at which the authors arrive:
Throughout the manuscript, demographic effects are either mentioned in passing or just ignored. These effects are important for understanding the population structure and genetic diversity of populations. Their environment/genetic diversity analysis is therefore flawed.
In the Conclusions, at line 349, the authors write, "The level of variation among populations is highly dependent on different environments." Perhaps they were speaking generally, but in the context of this experiment, the results do not indicate that genetic variation is highly determined by environmental conditions.

2. The strongest conclusion they arrive at is in relation to the lead author's 2019 paper. Starting on line 352, they state that north-south gene flow is greater than east-west gene flow. This is an unexpected result that deserves a more thorough discussion than it was given.

3. They also find that genetic diversity increases with elevated ploidy level, and that genetic distance is positively correlated with geographic distance. These are expected results, and the authors do a fair job of discussing them (excepting the fact that I was unable to discern which populations were which ploidy level).

4. The Supplementary Materials provided little in the way of underlying data. The raw microsatellite data was present with suitable metadata, but nothing else.

Additional comments

There are interesting angles in this manuscript that can be played up during revisions. The nuances of environmental variation across this longitudinal transect could be explored, along with better treatments of environment/diversity and/or environment/ploidy associations. Finding stronger population structure across longitude than latitude is an unexpected result that could be made exciting by tighter writing clearer statistical underpinnings. With a more careful and thorough look at the data and some additional analyses, this can be a very interesting and valuable paper.

·

Basic reporting

Listed in order of importance:
1. The flow cytometry data should be reported for each individual. This should include the genome size and inferred ploidy.
2. The English language should be improved to ensure that your international audience can clearly understand your text. Some examples of where the language could be improved include lines 61-64, 227-230, and 275-279. The current phrasing of these sections makes comprehension difficult.
3. I appreciate the inclusion of the genetic diversity values in tables 2 and 4, however the significance lettering in Table 4 is confusing. I recommend either describing the lettering in more detail in the table caption or not including the lettering and instead describing the significance in the text.
4. I thank you for providing figures that thoroughly describe your results, however figures 5 and 6 could be reduced and move to the supplementary material. STRUCTURE is a well-known algorithm therefore Figure 5 can be moved to the supplement. Additionally, Figure 6 can be reduced to only include the panels K=4 and K=8. All other panels in Figure 6 could be moved to the supplementary material.

Experimental design

no comment

Validity of the findings

Listed in order of importance:
1. In your analysis of genetic diversity in relation to polyploidy, individuals of the same ploidy level are treated as a population in order to compare genetic diversity among different ploidy levels (lines 128-129). In the discussion, you hypothesize “each chromosome replication process increased genetic diversity” (lines 303-304). This is a possibility, but it’s also possible that pentaploids and hexaploids have increased genetic diversity because their diploids and tetraploid progenitors were from C. dactylon populations with higher genetic diversity. This concept is discussed in Monnahan et al (2019) (https://doi.org/10.1101/411041). I recommend you either discuss this alternative hypothesis in your discussion or investigate this hypothesis in your data. You could test this alternative hypothesis by evaluating how the polyploids are distributed across the populations in relation to genetic diversity. For example, are pentaploids and hexaploids primarily found in populations with high diversity, such as Population 12?
2. Lines 268-272: You cite a study by Edillo et al. (2009) which found FST increases by 0.035 by for every 100-km. It is unclear whether you are suggesting their species also increases by 0.035 FST for every 100-km or whether they are using Edillo et al. as an example of isolation-by-distance. As Edillo et al. evaluates a different species, Culex pipiens, their FST calculations are not transmittable to your paper on Cynodon dactylon. These lines should be improved for clarity.
3. Lines 290-293: It is unclear if you are stating a new hypothesis or if this hypothesis has been supported in other studies. If this hypothesis has been previously acknowledged, please provide citations. If it is a new hypothesis, you should restate this sentence as a hypothesis.
4. Lines 232-233: Please consider exchanging “was beneficial to” with “was associated with”.

Additional comments

The methods and results of this study are well described, however portions of the introduction and discussion could be improved for clarity. I have noted these sections in my review. Additionally, I highly recommend the authors provide their flow cytometry data and investigate the relationship between ploidy distribution and genetic diversity.

Strengths of the paper:
-The EST-SSR data is very well documented and should serve as a good resource for future studies on this species.
-The study has a good sample size and marker density that is sufficient for a landscape genetics study.
-I commend the authors for the detailed description of their methods, particularly the genotyping methods. I feel the methods can be easily replicated if the flow cytometry data is included with the final manuscript.
-The research question is well defined (lines 84-85 and 91-95) and the authors have clearly answered each objective.
-The data on which the conclusions are based is statistically sound. I thank the authors for providing their results in well-formatted tables.
-I appreciate the discussion of the results of this study in relation to your previous study on C. dactylon across a latitudinal cline as it is highly relevant and contributes to the overall goal of your study.

---

## Round 0.2 · Major Revisions

Both the reviewers continue to have concerns about the clarity and readability of the writing. Reviewer 1 had some remaining concerns on methods and suggestions about modifying the Mantel test I agree with. I also agree with reviewer 2 about the necessity of including the information about ploidy variation within populations and correlations with longitude.

Reviewer 1 ·

Basic reporting

This article does not exhibit clear, unambiguous use of English. Though some grammatical, spelling, and punctuation errors have been fixed from the first version, many remain. Most of these errors are minor, but at times, ambiguous language prevented me from understanding parts of the experiment and the results. I will include below some of the errors and confusing/unclear segments I spotted:

50: "Gene" need not be capitalized.
57: These phrases are lifted from Zhang 2016. The changes the original authors' sentence are muddying; "mainly" has the wrong meaning for this context, and "populations" should be singular (the original sentence used the word "species," which can be singular or plural, but was singular in this case).
100: "include" should be "includes".
103: What "biological characteristics" are you referring to here?
104: What "obvious genetic pattern" are you referring to here? Genetic structure?
105: These Objectives have no verb, consider something like "quantify genetic diversity", "test the relationship", etc.
109: This sentence is confusing. You just finished laying out the objectives of the study, and you more directly into stating what the results do. Also, what does "broaden the genetic base" mean? Shouldn't "understand" be "understanding"? More generally, by this point in the introduction, you shouldn't still be explaining the design of the experiment ("by tapping into the gene pools of the wild C. dactylon along longitudinal gradients"); that should be sufficiently covered prior to stating the objectives.
143: "Calculating" seems like the wrong word here.
144: "The variation degree" seems wrong. Maybe "the degree of variation"?
147: The referencing of the method of Zhang 2020 is not proper. Should be something like, "based on the method outlined by Zhang et al (2020).".
151: "test Mantel test" seems wrong. Also, some clarification would be helpful here. What matrices are being compared, and how are they computed?
188: The words "significant" and "significance" are misspelled throughout the manuscript.
192: "had the higher" should be "had higher".
194: "was", not "were".
212: Sentence is missing a word or two. "was identified", not "were identified".
214, 223: "Populations", not "Population".
222: Sentence is incomplete.
229: "clusters", not "cluster".
231-241: You now mention populations with "No.", but in the previous paragraph you did not. Be consistent with your naming conventions. Also, see note from 214.
234: "Each of the other", not "Each of other".
244: "Those prediction was" is improper.
283: "exist", not "exit".
285. Where is the "on the other hand"? Also, "genotypes", not "genotype". Also, missing parenthesis.
334: This sentence is confusing. How does this grass directly emerge from sites of low and high longitude?

The structure of the article is appropriate. The figures and tables that were included were relevant, and easy to read. I did note several places were the text unclear for reasons related to narrative flow or lack of explanation:

166: K value selection should be discussed after introduction of STRUCTURE and the permutation test.
238: This sentence may be better placed in the Discussion.
267-270: These sentences seem to be placed here randomly. What is the logical narrative flow?
275: Where does this come from?
281-305: This paragraph is difficult to follow. The first sentence should do a better job of setting up the main idea of the paragraph, and the following sentences need to support that idea. Cutting down on the number of referenced sources here may help.
335-360: Would you expect that the mid-longitude populations, rising from complex hybridizations, might have elevated ploidy levels? Was this investigated or tested?
364: From where are you inferring that one is beneficial to the other, rather than a simple correlation?
369: I recommend a new paragraph here.

Though a good amount of literature is cited, in my opinion, it is not properly used. In numerous instances, the authors' wording is similar enough to the reference material that it makes me uncomfortable. Though it is clear that efforts were made to adapt the words of others into new phrases, in my opinion, a greater effort must yet be made.

I feel that this is a coherent body of work, but based on the authors' response to my first review, I also suspect that the authors are holding back some data and analysis (the spatial patterns of ploidy levels) for a later publication. I don't know if I am right, and if I am, I do not speculate as to their motives. I do think that that data would contribute meaningfully to this article, were they to have included it.

Experimental design

This article qualifies as primary research which falls within the aims and scope of this journal. The research questions are defined and relevant, though I am still unsure why so much focus is placed on longitude. On lines 61-66, longitude is shown to be an important variable because of the other things (MAT, MAR, altitude, migration barriers) that vary across it. Once again, I am led to ask, why does longitude itself matter, other than as a proxy for other biologically-relevant variables or for distance in general? Either important elements of the environment vary across longitude, in which case those elements can be discussed more directly, or longitude is a benign measure with no discernable biologically meaningful factors that vary across it, in which case it can be replaced with geographic distance, especially in the case of this experiment, in which almost all geographic distance between samples is across longitude.

The methods were mostly thorough, though there were sections that were not detailed enough, and others that were absent. I could not deduce the methods by which the authors arrived at a number of their results, and there was a time or two that I wished there were an additional figure or table. See below:

155: Mantel test is mentioned here again. On line 151, this test is between genetic diversity (genetic distance?) and environmental factors (environmental distance?). On line 155, this test is between genetic distance and geographical distance. Why not combine the descriptions of these Mantel tests into the same paragraph, or at least in the same section of the Methods?
166: Consider mention of Delta K, to which you dedicate one of your supplementary figures.
177: What is the value of g that you used? Also, my understanding is that Cynodon dactylon is primarily rhizomatous, reproducing asexually. If I'm right, how does that affect the use of this statistic, if at all?
194: What test is this referring to? In this paragraph, you have been plotting genetic diversity (He and I) across longitude, as in Figure 2 and Table 2. Have you also plotted these diversity metrics across MAF and MAT, or run a simple Pearson correlation? Why not include a table or figure, or reference to methods? Also, did you hypothesize that higher/lower MAF and/or MAT would increase/decrease the level of genetic diversity in these populations (for example, higher temperature and rainfall result in higher genetic diversity)?
205: I cannot find the methods that outline how you calculated gene flow. If you used FST to estimate it, then your estimate may be inaccurate, as your longitudinal sampling likely violates the island biogeographic model (Whitlock, Michael C., and David E. Mccauley. "Indirect measures of gene flow and migration: FST≠ 1/(4Nm+ 1)." Heredity 82.2 (1999): 117-125.).

Validity of the findings

Since the first version, efforts have been made to discuss the demographic history of these populations. Their use of the mismatch distribution and neutrality tests is an improvement.

Another improvement is their incorporation of the Mantel test. However, the authors may note that a Mantel test is still another application of a correlation statistic (between pairwise distances rather than between the values themselves). Furthermore, this analysis still does not account for the effects of demography. Had demographic effects been accounted for, stronger relationships between genetic and environmental distances may have been uncovered, while other noted correlations may be explained by neutral processes.

The Mantel test analysis can be improved. Looking specifically at 200-204: Mantel finds positive correlations between genetic, geographic, and precipitation distance. It is unclear whether geographic or precipitation distance is the stronger determinant of genetic distance. It is clear from the language of the sentence at 202 that the authors believe geographic distance is the determinant, and precipitation is strongly correlated with genetic distance only because precipitation distance is also strongly correlated with geographic distance. This hypothesis could be tested by way of a partial Mantel test: Test the correlation between genetic distance and geographic distance, while controlling for precipitation distance. This could also be reversed, testing the correlation between genetic and precipitation distance while controlling for geographic distance. This would disentangle these relationships and allow clearer interpretation, greatly improving this analysis.

In the cases where methods were insufficiently detailed, I am uncertain whether those results are valid or not. This mostly applies to their estimation of gene flow.

The final paragraphs of the paper attempt to rationalize unexpected results by speculating human interference. The authors properly identify this speculation when doing so, and do not engage in much speculation beyond this context.

Additional comments

The writing in the Discussion (and to a lesser degree, the Introduction) comes across as disjointed and lacking in narrative direction. I suspect that many of the sentences that the authors use in these sections are taken from their referenced sources with only minor modifications. The end result of writing in this way is that each sentence feels like a discrete idea in a long chain, isolated by conceptual distance from the others. Investing effort into greater synthesis of these ideas will improve the clarity of the story you are presenting in this article.

·

Basic reporting

1. Genome size and inferred ploidy has not been provided. Without this information, analyses investigating the relationship between ploidy and genetic diversity cannot be replicate. The submission of this raw data is necessary.

2. The English language should be improved to ensure that your international audience can clearly understand your text. The text is greatly improved from the first version although significant improvement is needed in the Result and Discussion. Some examples of where the language could be improved include lines 266-274, 301-305, and 325-334. Additionally, lines 202-203 are unclear. Is the correlation between MAP and genetic distance dependent on the correlation between geography and MAP? Or did you run separate Mantel tests for both MAP and geographic distance?

3. Table 4 is not referenced and should be move to the supplement. It is improperly referenced on line 243 when Table 5 should instead be referenced.

Experimental design

no comment

Validity of the findings

In their rebuttal, the author's clarified, "Composition of different ploidy levels within a population at different longitudes was almost similar in another unpublished paper. Longitude was not highly correlated with ploidy level". The paper would benefit from including this analysis as it's currently unclear how the ploidy levels are structured across the populations. This information clarifies that the increasing genetic diversity with ploidy is likely due to the polyploid event rather than the pentaploids and hexaploids originated from populations of higher diversity.

Additional comments

The first revision of this article is a significant improvement from the previous version, however portions of the Results and Discussion should be improved for clarity. I have noted these sections in my review. Additionally, the raw flow cytometry data and inferred ploidy levels are needed for the analysis of genetic diversity across ploidy levels to be replicable.

---

## Round 0.3 · Major Revisions

Thank you for the improvements to the manuscript, but both reviewers have additional comments that should be carefully considered. In particular, why not include the ploidy analysis suggested? Both also point out difficulties with the writing.

I note that I am very unlikely to send the paper back to reviewers; a revision that does not seriously address and make notable changes in response to this round will likely be rejected.

Reviewer 1 ·

Basic reporting

This third version of the manuscript is a marked improvement, particularly with regards to the writing. Clearly, efforts were made to improve the clarity of the ideas. There is also a reduction in the number of sentences lifted with improper/minor/insufficient modification from other published sources, which also had the benefit of improving the narrative flow of the manuscript. Still, the paper abounds with grammatical, spelling, punctuation, and word choice errors, some of which are new to this third version of the manuscript.

Your inclusion of the raw data as requested by Reviewer 2 is a valuable addition to the manuscript, but its value is limited somewhat by the refusal to carry out certain analyses with it. I agree with Reviewer 2's assessment that this manuscript would be strengthened by an analysis showing whether ploidy level is correlated with longitude, but you have elected instead to analyze this data in a separate paper. I am left to believe that you excluded this analysis to increase publication count.

Experimental design

The methods section is more complete and has greater detail. Recommended analyses by myself and Reviewer 2 were not carried out, in particular the ploidy level/longitude analysis. I also suggested the addition of a partial Mantel test (test the correlation between genetic distance and another distance matrix, while controlling for other distance matrices), but instead, you cut the sentence from the Results section that prompted this suggestion.

In my second review of your manuscript, I again attempted to explain my problem with the idea of longitude being a meaningful environmental gradient around which to serve as the basis for your manuscript. I would like to address your rebuttal to this note. Starting with the second paragraph of your rebuttal to this section, you clarified your methodology of using Mantel tests between genetic diversity and precipitation, temperature, and geographic distances. You noted that precipitation, temperature, and geographic distance all vary across longitude. This was my point as well, and it does not bolster the importance of longitude. Your testing of the effects of precipitation, temperature, and geographic distance is appropriate, and makes sense, yet throughout the manuscript, it is longitude that is discussed as a gradient of ecological importance. This paragraph does not make a case for longitude as a biologically-relevant variable.

Validity of the findings

I have no additional comments.

Additional comments

I have an additional note to make regarding your response to my question regarding the relevance of longitude as an environmental variable. In the first paragraph of your response, you stated:

"Understanding the population genetic pattern and process of gene flow requires a detailed knowledge of how landscape characteristics structure populations. Higher temporal resolution can be provided using population genetic sampling and measures for detecting genetic relationships within continuously distributed populations in a landscape gradient (Landguth et al., 2010; Blair et al., 2012)."

I had difficulty understanding these sentences. What is the population genetic pattern of gene flow? How does temporal resolution fit into this question? Why are Landguth 2010 and Blair 2012 cited here, but not in your manuscript? After looking around online, I came across Manel 2003, "Landscape genetics: combining landscape ecology and population genetics". This is the first sentence of the abstract of that paper:

"Understanding the processes and patterns of gene flow and local adaptation requires a detailed knowledge of how landscape characteristics structure populations."

The next two sentences are from Bolliger 2014, "Landscape genetics since 2003: status, challenges and future directions":

"These individual-based methods provide much higher temporal resolution for detecting landscape genetic relationships (Landguth et al. 2010; Blair et al. 2012). This makes individual-based approaches particularly valuable for analyses within continuously distributed populations, or in gradient landscapes."

It became clear that your rebuttal paragraph was pieced together from the sentences of other published literature. The effort to synthesize this content into your own words was insufficient, leading to a puzzling paragraph that failed to explain or support your case. The sentences were similar enough to their sources that they were able to be quickly tracked down to their sources, despite being uncredited. You cited neither Manel 2003 nor Bolliger 2014, the sources from which you drew these segments, yet cited Landguth 2010 and Blair 2012 because Bolliger 2014 did.

I observed this style of writing in the prior version of the manuscript, and commented on it in my second review. You replied in your rebuttal that the whole manuscript had been carefully checked and revised, but I have doubts that the manuscript is truly free of improper incorporation of the unacknowledged work of others. I leave it to the editors to determine if this is the case.

·

Basic reporting

1. Although the language has been greatly improved, there are still many typos and grammatical errors that must be resolved to improve clarity. Lines that require improvement include:
52: What are you “describing the evolutionary” significance of?
58-65: These sentences are confusing. You begin by saying longitude is the driving factor of some species distributions and then explain this is only because specific environmental variables are correlated with longitude. These sentences can be improved by first emphasizing that longitude is correlated with environmental variables and then continuing with examples.
87 and 89: “is” should be “are”
248: Which population?
251: I believe the authors mean “positive” rather than “negative” according to Table 4.
258, 360 and 405: “exited” should be “exists”
259: “differences in” should be added after “significant”
298-299: This sentence is unclear.
310: Currently phrasing suggests higher FST is the cause of local genetic differentiation. This should be rephrased to state higher FST indicates differentiation.
379-382: This sentence is unclear.
387-388: “because longitude might be easily affected by ecological isolation” should be replaced with “because populations were ecologically isolated”.

2. 330-332: McMillian’s theories apply to North America. How does this apply to your region of study?

Experimental design

1. 197-200: Why were the populations grouped in this manner? This STRUCTURE and UPGMA results suggest these are not genetic groups therefore it’s unclear what their comparison accomplishes.
If these groups have biological meaning, how were the genetic diversity parameters calculated for each group?

2. 255-261: The authors should include the p-values for comparisons made.

3. I agree with Reviewer 1’s previous review that a mantel test should be conducted between genetic distance and geographic controlling for precipitation distance in order to test the author’s hypothesis that geographic distance determines genetic differentiation.

Validity of the findings

1. 284: The authors argue that higher genetic diversity of populations at mid-longitudes could result from higher “communication” between populations. This is contradictory to the low levels of gene flow reported at line 215.

2. 289-291: The authors argue that the PCoA and Mantel tests show genetic clustering occurs along the longitudinal gradient yet the PCoA shows populations from across the entire cline form clusters rather than with populations geographically nearby. The clustering in the PCoA is contradicted by the positive correlation between genetic distance and geographical distance. As reviewer 1 suggested, this analysis should be re-run controlling for precipitation distance to disentangle this relationship.

3. 391-394: The relationship between ploidy, genetic diversity, and longitude were not tested. The authors only tested the relationship between ploidy and genetic diversity. The phrasing of this sentence should be improved to clarify that diversity increases with ploidy, not ploidy across longitude.

---

## Round 0.4 · accepted · Accept

Thank you for your careful response to reviews!